# Silicon Modulates Molecular and Physiological Activities in *Lsi1* Transgenic and Wild Lemont Rice Seedlings under Arsenic Stress

**Mohammad Reza Boorboori** [ID]**, Wenxiong Lin, Yanyang Jiao and Changxun Fang** *[ID]

Fujian Provincial Key Laboratory of Agroecological Processing and Safety Monitoring, College of Life Sciences, Fujian Agriculture and Forestry University, Fuzhou 350002, China; m.boorboori@yahoo.com (M.R.B.); wenxiong181@163.com (W.L.); daniel2zhu@163.com (Y.J.)
* Correspondence: fcx007@fafu.edu.cn

**Abstract:** Arsenic is one of the most dangerous metalloids, and silicon is a helpful element supporting plants to withstand stress. In this study, three factors were considered, including rice accessions with three different lines, including *Lsi1*-RNAi line (LE-R), *Lsi1* overexpression line (LE-OE), and their wild type (LE-WT), and silicon and arsenic treatments with two different levels. Analysis of variance in dry weight biomass, protein content, arsenic, and silicon concentration has shown a significant interaction between three factors. Further analysis showed that the silicon concentration of all rice seedlings under silicon treatments increased significantly. The LE-OE line has shown a higher ability to absorb silicon in hydroponic conditions than the wild type, and when the seedlings were exposed to arsenic, the concentration of arsenic in all lines increased significantly. Adding silicon to over-expressed rice lines with the Lsi1 gene creates better arsenic resistance than their wild type. These findings confirmed antagonism between silicon and arsenic, and seedlings exposed to arsenic showed a reduction in silicon concentration in all rice lines. RNA-seq analysis showed 106 differentially expressed genes in the LE-OE line, including 75 up-regulated genes and 31 down-regulated genes. DEGs in the LE-R line were 449 genes, including 190 up-regulated and 259 down-regulated genes. Adding treatment has changed the expression of Calcium-binding EGF domain-containing, Os10g0530500, Os05g0240200 in both LE-OE and LE-R roots. They showed that transgenic cultivars were more resistant to arsenic than wild-type, especially when silicon was added to the culture medium.

**Keywords:** *Oryza sativa* L.; arsenic and silicon antagonism; protein content; RNA-seq; antioxidative enzyme genes

## 1. Introduction

The effects of environmental pollution on nature and humans have been increasingly recognized over the past decades. One of the most imminent threats to nature and ecosystems is water and soil pollution by metalloids and heavy metals. They penetrate biological cycles, cause fundamental changes in ecosystems, and have harmful environmental effects [1]. Arsenic (As) is a toxic metalloid with different oxidation numbers (−3, +3, +5). However, it occurs naturally in the soil in low concentrations, mainly originating from human activities [2]. Humans are mainly exposed to As by consuming contaminated foods and water in areas contaminated by mining activities [3]. Arsenic disrupts blood circulation, damages the gastrointestinal tract and nervous system, and damages the skin; therefore, it threatens human health, and its acute poisoning can be fatal [4]. Arsenic compounds penetrate the environment through many materials used in industry; this element is used significantly as herbicides, fungicides, insecticides, algaecides, etc. [5].

Silicon (Si) is a fundamental element in the soil and the earth's crust [6,7], and it contains 1 to 45% of the dry weight of the soil, but only 0.1 to 0.6% of it is soluble [8]. Si is an immobile element in plants, but it is not an essential element. However, many plants,

including rice, need Si for normal growth, especially in stress conditions that can increase plant resistance [7,9]. Plants absorb Si as $Si(OH)_3O^-$ or $Si(OH)_4$ (Orthosilicic acid), and it makes up 0.1 to 10% of the plant's dry weight. Silicon also increases the resistance of the plant to non-biotic and biotic stresses [10].

Rice is an essential crop in the world that provides 40% of people's food. It provides two thirds of the calories needed by two billion people in Asia and is also the main source of protein for this population [11]. As rice is a stacked Si plant, it is often used as a model plant for silicon uptake, transport, and accumulation [12]. However, adding Si to soil and water significantly reduces the concentration of As in rice. Some reports have shown that Si can contribute to methylation growth in rice tissues and affect rice accumulation of different As species [13].

Moreover, Si decreases As uptake and transport in plants and reduces the As amount in polished and brown rice to 33% and 22%, respectively [14]. Ali, Isayenkov [15] reported that silicon could reduce arsenic transfer in rice. Thus, it reduced the concentration of As in straw and leaves by 50%. Reports have indicated that different rice genotypes also have different levels of As storage capacity [16,17], and various experiments have shown significant differences in the ability of different types of rice in arsenic accumulation in straw and grain [2,18].

Two silicon carriers in rice (*Lsi1* and *Lsi2*) have reportedly facilitated arsenic transport from soil to plants [19]. *Lsi1* is an invasive transporter responsible for transporting silicon and arsenic (III) from culture solutions to rice root cells. Therefore, arsenic transfer from the environment to the plant is mediated by *Lsi1* [20]. Research has shown that the *Lsi1* expression gene in rice increases plant resistance to As (III) treatment [21].

Plants have different mechanisms for detoxifying As, one of which is the production of compounds between As and thiols, same as glutathione, which reduces the motility of As in vacuoles [22]. Detoxification of reactive oxygen species (ROS) is achieved by synthesizing antioxidants like glutathione S-transferase, superoxide dismutase, catalase, and glutathione [23].

The present study attempts to elucidate *Lsi1* mechanisms regulating Si uptake to affect arsenic accumulation in rice seedlings. We hypothesized that detoxification-related genes were involved in regulating As tolerance in rice. To detect it, three different rice lines, including wild Lemont, LE-OE, and LE-R, where the level of *Lsi1* transcript was different, were used to study the transcription of As and Si genes in these lines under arsenic stress.

## 2. Materials and Methods

### 2.1. Planting Conditions

In the present study, Lemont rice (*Oryza sativa* L. subsp. Japonica) and its transgenic lines with *Lsi1*-RNA interference (*Lsi1*-RNAi) and overexpression of *Lsi1* (*Lsi1*-OE) were used. Lemont rice was stored in the agricultural processing and safety monitoring main laboratory of Fujian University of Agriculture and Forestry (Fuzhou, China), and the *Lsi1* Lemont transgenic lines were germinated in the previous studies. [24]. Seeds of different types of rice, for example, wild Lemont (LE-WT) and *Lsi1*-OE (LE-OE) and *Lsi1*-RNAi (LE-R) lines, were prepared and disinfected using alcohol and sodium hypochlorite. For germination, the seeds were kept in a damp paper towel in the culture chamber. After one week, the germinated seeds were transferred to pots (2.5 L) containing the culture solution, and the culture solution was prepared, according to Cock, Yoshida [25]. The nutrient solution included: $(NH_4)2SO_4$ (48.2 mg $L^{-1}$), $Ca(NO_3)_2 \cdot 4H_2O$ (86.43 mg $L^{-1}$), $K_2SO_4$ (14.9 mg $L^{-1}$), $Na_2SiO_4 \cdot 9H_2O$ (200 mg $L^{-1}$), $KNO_3$ (18.5 mg $L^{-1}$), $FeSO_4 \cdot 7H_2O$ (45.7 mg $L^{-1}$), $H_3BO_3$ (1.43 mg $L^{-1}$), EDTA (48.44 mg $L^{-1}$), $CuSO_4 \cdot 5H_2O$ (0.04 mg $L^{-1}$), $KH_2PO_4$ (24.8 mg $L^{-1}$), $MnCl_2 \cdot 4H_2O$ (0.905 mg $L^{-1}$), $Na_2MoO_4 \cdot 2H_2O$ (0.045 mg $L^{-1}$), $ZnSO_4 \cdot 7H_2O$ (0.11 mg $L^{-1}$), and $MgSO_4 \cdot 7H_2O$ (135.06 mg $L^{-1}$). The acidity of the pot solution was adjusted daily to pH 5.8, and, at the end of each week, the pot solution was replaced.

Treatments began when the rice seedlings had three leaves. In a complete factorial design, we exposed the seedlings to two Si levels ($Na_2SiO_3 \cdot 9H_2O$) (0 and 0.70 mM) and two As (III) levels ($NaAsO_2$) (0 and 30 µM). Thus, we had four different treatments: control (I), 0.70 mM Si (II), 30 µM As (III), 30 µM As + 0.70 mM Si (IV). NaCl equivalent was added to the solution to compensate for the sodium amount added to the culture medium in the form of $Na_2SiO_4 \cdot 9H_2O$. The experiment was conducted in a completely randomized design with three replications, and random rice samples were selected one and two weeks after the start of treatment. The roots and shoots were cleaned in distilled water, and the remaining residue in the root zone was washed with distilled water in 0.5 mM $CaCl_2$ solution for 30 min. After washing, all samples were transferred to the freezer at $-10\ °C$ and stored until enzyme activities were determined.

For Transcriptomic analyses and quantitative RT-PCR, rice seedlings containing three leaves were treated with 30 µM As + 0.70 mM Si for three days. After three days, all seedlings were harvested, and the shoots and roots were separated, instantly immersed in liquid nitrogen, and stored rapidly at -80 °C. In this experiment, we used LE-WT as CK and LE-OE and LE-R as treatments.

### 2.2. As Concentration Measurement

We used the method of Meharg and Jardine [26] for determining As concentration in rice samples. To each 0.2 g of dry samples, 1 mL of condensed nitric acid was added. After the mixture had been stored at room temperature for 24 h, 1 mL of oxygenated water was added. Subsequently, the test tubes were placed at 70 °C for half an hour. Then they were transferred to a temperature of 100 °C for one hour until all nitric acid from the samples had vaporized. After cooling, the obtained solution was filtered, and distilled water was added until the volume reached 50 mL. Then, 10% chloric acid (5 mL), 10% potassium iodide (10 mL), and 5% ascorbic acid (5 mL) were added to 1 mL of the diluted solution of each sample. Arsenic concentration was finally measured using atomic absorption spectrometry (Shimadzu 6200) with a hydride production apparatus (FIG 6200).

### 2.3. Si Concentration and Dry Weight Measurement

The samples were ground and put into an oven at 70 °C for two days. Then, dry samples (0.5 g) were microwave-digested in the mixture of 62% $HNO_3$ (3 mL), 30% $H_2O_2$ (3 mL), and 46% HF (2 mL). The digested samples were diluted with boric acid (4%) to 100 mL. We used a colorimetric molybdenum blue method to measure Si concentration in the digest solution [27]. To sample (0.2 mL) $H_2O$ (2.7 mL), by10% $(NH_4)_6 Mo_7O_2$ (0.2 mL), 0.2% NaCl (1.5 mL), 20% tartaric acid (0.2 mL), and then reduced agent (0.2 mL) were added. We prepared a reducing agent by dissolving 1-amino-2-naphthol- 4-sulfonic acid (0.5 g), $Na_2SO_3$ (1 g), and $NaHSO_3$ (30 g) in the water (200 mL). We measured the absorbance at 600 nm after 1 h. A series of Na2SiO₃ solutions along a concentration gradient was used to create the standard curve.

### 2.4. Soluble Protein Contents Measurement

The amount of total soluble protein was measured using the Bradford [28] method. Samples of plant material are powdered with liquid nitrogen using precooled mortars and pestles. Each sample supernatant and working standard solution were transferred to assay tubes. A blank containing 20 µL extraction buffer was also prepared. The supernatants (20 µL) were mixed with 1 mL of Bradford solution. An amount of 100 mg Coomassie Brilliant Blue G-250 in 50 mL 95% ethanol was dissolved and 100 mL 85% (*w/v*) phosphoric acid added. Once the dye had completely dissolved, it was diluted to 1 L with deionized water and, in five minutes, the mixture absorbance was measured at 595. The protein amount was quantified by placing the number onto the standard curve.

### 2.5. Transcriptomic Analyses

Total RNA was extracted from rice roots by using TRIzol (Invitrogen) according to the manufacturer's instructions [29]. RNA quality was measured on an Agilent 2100 Bio-analyzer system (Agilent Technologies, Santa Clara, CA, USA). Total RNA (0.75 µg) from each test sample was treated with oligo (dT) beads to enrich mRNA and fragmented using fragmentation buffer (New England Biolabs Ltd., Hitchin, UK). mRNA was reverse-transcribed into cDNA using random hexamers, and double-stranded cDNA (ds cDNA) was synthesized and purified. After filling in the cohesive ends to yield blunt ends, ds cDNA was combined with poly (A) and ligated to Illumina sequencing adapters (Illumina Inc., San Diego, CA, USA). The ligation products were size-selected by AMpure XP beads (Beckman Coulter Life Science) and amplified by PCR, and cDNA libraries were sequenced on an Illumina Hiseq PE150 at Allwegene Tech. (China). Differential expression analysis (DEG) was done by using the DESeq R (1.10.1) package. To check the false discovery rate (FDR), we obtained the *p*-values via Benjamini and Hochberg's method. Significant false genes were detected by values < 0.05 [30]. For GO (gene ontology) enrichment analysis, a single enrichment analysis tool (SEA) was done on The Viral Proteomic Tree Server (ViPTree), with default parameters set, and an adjusted FDR threshold was set to $p < 0.05$.

### 2.6. Quantitative RT-PCR

RNA was extracted from the roots using the Trizol method with slight modification [29]. We used one µg total RNA for the first-strand cDNA synthesis, according to the instructions of the TransScript One-Step gDNA Removal and cDNA Synthesis SuperMix manufacturer (TransGen Biotech Co., Ltd. Beijing, China). Relative levels of GST-related genes were increased rice plant resistance to arsenic stress, and they were determined through quantitative RT-PCR in three replications. The sequences of primer are specified in Table 1.

**Table 1.** Primers of gene used for qRT-PCR in this study.

| Gene | Primer Sequence (5′-3′) | Primer Length (nt) | Tm (°C) | PCR Product Length (bp) |
|---|---|---|---|---|
| Os10g0530500-F | ACAACATGTTCCCTGGAATGG | 21 | 55.6 | 166 |
| Os10g0530500-R | TCGACGTACCCGATGGAGTC | 20 | 59.5 | |
| Os08g0522400-F | CAAAGACAAGCTTTCACCGTAA | 22 | 53.9 | 282 |
| Os08g0522400-R | CAGAAAAGAACGCTGCCTTTAA | 22 | 53.9 | |
| Os01g0369700-F | GTTCGGTGAGATTCCAGTACTG | 22 | 57.7 | 86 |
| Os01g0369700-R | TGTACTTGCGAAAGATGTACCT | 22 | 53.9 | |
| GSTU1-F | GTGAGTTTGTTGTTACCGTTGA | 22 | 53.9 | 84 |
| GSTU1-R | TGACAATCTCAGAGAATCGGAG | 22 | 55.8 | |
| GSTZ5-F | AAGATTGTCGCGATTGATCTTG | 22 | 53.9 | 95 |
| GSTZ5-R | TGATTGTTGTGCTCAAGTGAAG | 22 | 53.9 | |
| Os05g0240200-F | CTGAAGATGTTGGCTACTTTCG | 22 | 55.8 | 177 |
| Os05g0240200-R | CATCTTTCAGGAACCGCATATG | 22 | 55.8 | |
| ATG8D-F | TCTTCTGGAGTCTACACGTCTA | 22 | 55.8 | 111 |
| ATG8D-R | GTCTTCTTCCTTGATGCGAATC | 22 | 55.8 | |
| LOC_Os10g10130-F | TTTGGATGCTGTCTTGAAACTG | 22 | 53.9 | 87 |
| LOC_Os10g10130-R | ATCTGGATGAAGTAGTCCGAAC | 22 | 55.8 | |
| Os02g0685200-F | CGGTGGGTTCTCGAATAACTC | 21 | 57.6 | 193 |
| Os02g0685200-R | CGTGGTTGCAATTGACATCTTA | 22 | 53.9 | |
| actin1(*Os03g0718100*)-F | CTTCATAGGAATGGAAGCTGCGGGTA | 26 | 61.2 | 197 |
| actin1(*Os03g0718100*)-R | CGACCACCTTGATCTTCATGCTGCTA | 26 | 61.2 | |

Actin-1(*Os03g0718100*) was used as a reference gene. The primers and probes are designed for Quantitative RT-PCR analysis by using online tools at https://biodb.swu.edu.cn/qprimerdb/. The quantitative RT-PCR reaction system was prepared using TransStart Tip Green qPCR SuperMix and an Eppendorf realplex4 instrument. The reaction process was as follows: predenaturation at 94 °C for 30 s, denaturation at 94 °C for 5 s, annealing at 53 °C for 15 s, extension at 72 °C for 10 s; 42 cycles. When the amplification was finished, the melting curve analysis was conducted, and the specificity of the product was determined based on the melting curve (Figures 1 and 2). Each candidate mRNA was set with four independent replicates. The relative expression of the gene was calculated by the $2^{-\triangle\triangle Ct}$ method with the threshold cycle values (Ct) of each candidate mRNA in both the control and test samples [31].

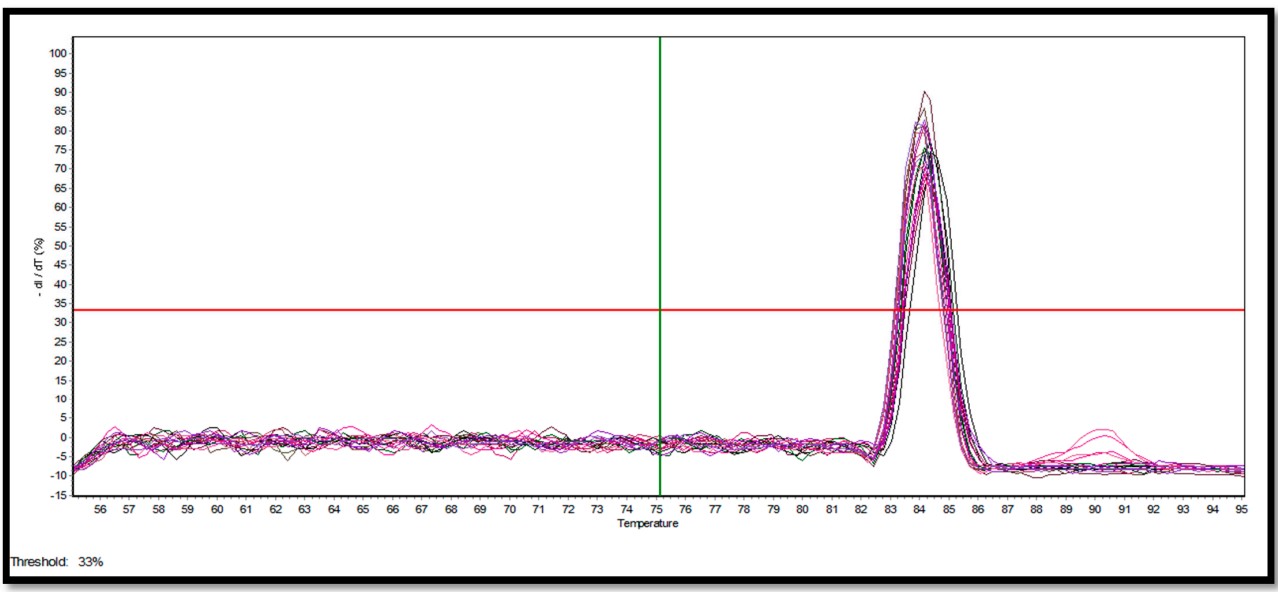

**Figure 1.** The melt curve of the amplified fragment of the *Actin-1*(*Os03g0718100*) gene.

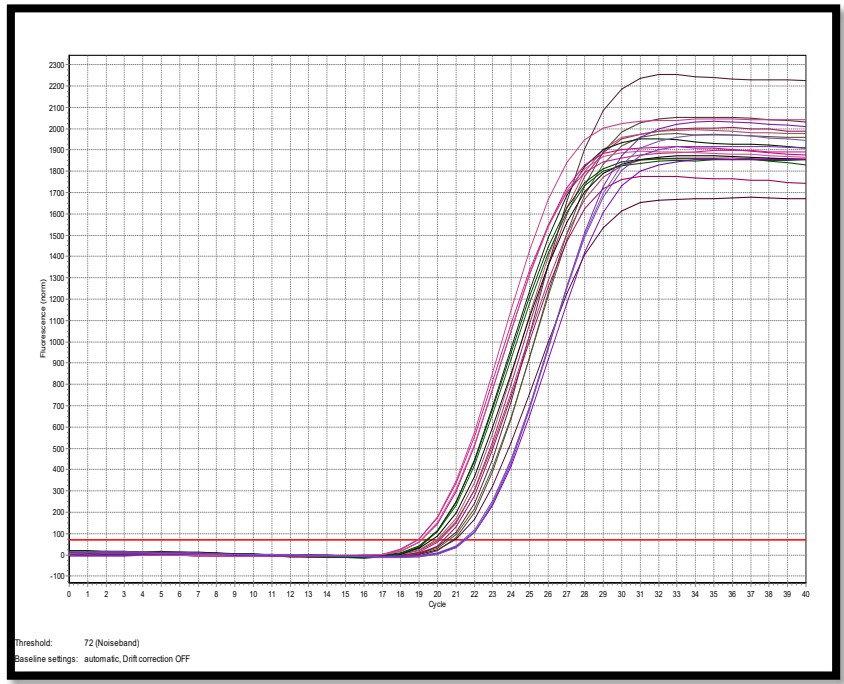

**Figure 2.** Amplification curve of the *Actin-1*(*Os03g0718100*) gene.

### 2.7. Statistical Analyses

This experiment was organized in a completely randomized factorial design and three replications. Total determination was statistically analyzed using one-way ANOVA with pairwise LSD. tests ($p \leq 0.05$). Analysis was conducted using SPSS 26.0 software (IBM Corp., Armonk, NY, USA).

## 3. Results

### 3.1. The Role of Si and As Treatments on As Concentration of Rice Shoots and Roots

Data analysis showed a significant interaction between As × Si × lines in rice under different treatments (Table 2). When we increased As in the culture medium, the accumulations of arsenic in different rice tissues increased, and the results showed more As in roots than shoots of different rice (Figure 3A,B). Arsenic treatment significantly increased As concentration ($p \leq 0.05$) in rice shoots compared to CK The highest increase was seen in LE-WT shoots in the first and second weeks (30 μM As), with 5.29 mg kg$^{-1}$ DW and 9.72 mg kg$^{-1}$ DW, As concentration, respectively. The concentration of As in different seedlings treated with Si was 0 mg kg$^{-1}$ DW in both weeks. As + Si treatment was decreased As concentration in different rice shoots compared to As treatment only; and the most considerable decrease saw in LE-OE shoots in both weeks (2.32 mg kg$^{-1}$ DW, and 2.37 mg kg$^{-1}$ DW, respectively) (Figure 3A).

**Table 2.** Results (F value) of ANOVA on effects of line, As, Si, and interactions among them on As concentration and dry weight of shoot and root of Lemont rice lines at different sampling times.

| Sampling Time | Parameter | Line | As | Si | Line × As | Line × Si | As × Si | Line × As × Si |
|---|---|---|---|---|---|---|---|---|
| 1st week | Shoot As concentration | 344.84 ** | 15,523.48 ** | 2192.96 ** | 236.03 ** | 85.79 ** | 848.92 ** | 151.63 ** |
| | Root As concentration | 182.45 ** | 25,123.40 ** | 5047.21 ** | 122.71 ** | 270.55 ** | 1568.16 ** | 213.83 ** |
| | Shoot dry weight | 6.96 ** | 69.76 ** | 34.11 ** | 0.32 ns | 0.55 ns | 1.05 ns | 0.32 ns |
| | Root dry weight | 3.29 * | 62.42 ** | 35.89 ** | 0.37 ns | 0.33 ns | 1.90 ns | 0.09 ns |
| 2nd week | Shoot As concentration | 987.26 ** | 14,477.16 ** | 3755.15 ** | 861.59 ** | 359.47 ** | 1547.55 ** | 299.44 ** |
| | Root As concentration | 395.65 ** | 28,989.96 ** | 7140.00 ** | 244.32 ** | 273.65 ** | 1789.42 ** | 122.37 ** |
| | Shoot dry weight | 3.98 * | 77.01 ** | 41.90 ** | 0.17 ns | 0.29 ns | 2.22 ns | 0.16 ns |
| | Root dry weight | 1.69 ns | 74.41 ** | 47.70 ** | 0.12 ns | 0.28 ns | 2.23 ns | 0.11 ns |

** Significant at the level of 1%, * significant at the level of 5%, ns nonsignificant.

By adding As to the culture solution, the concentration of root arsenic increased compared to CK ($p \leq 0.05$), the highest As concentration was observed in LE-WT roots in the first week, but in LE-R roots in the second weeks (Figure 3B). The highest concentrations of arsenic in the first and second weeks were 256.8 mg kg$^{-1}$ DW and 376.08 mg kg$^{-1}$ DW. The arsenic concentration of Si-treated roots was 0 mg kg$^{-1}$ DW every two weeks. As concentration in rice seedlings treated with As + Si decreased compared to As only exposure, the lowest arsenic concentration was observed in the first and second weeks in LE-WT roots, 48.41 mg kg$^{-1}$ DW and 67.98 mg kg$^{-1}$ DW, respectively.

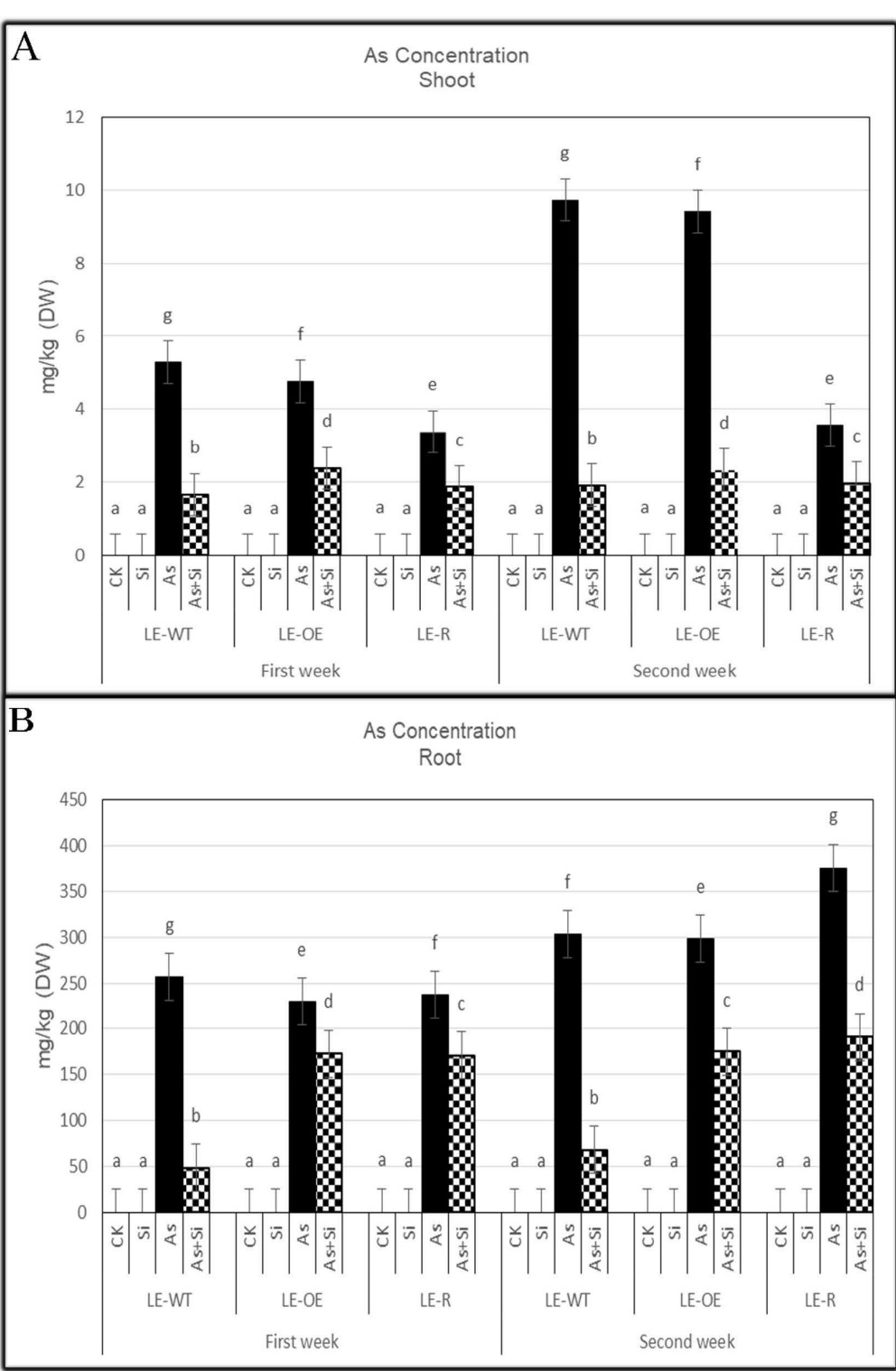

**Figure 3.** As concentration in shoots (**A**) and roots (**B**) of LE-WT, LE-OE, and LE-R first and second weeks after adding different treatments. The different letters on the bars significantly showed differences ($p \leq 0.05$) between treatments, respectively (DW: dry weight).

### 3.2. The Role of Si and As Treatments on Si Concentration of Rice Shoots and Roots

Figure 4 determined Si concentration in the shoots and roots of three rice lines under different treatments and also showed the Lsi1 gene increase the capacity of different tissues for further Si accumulation.

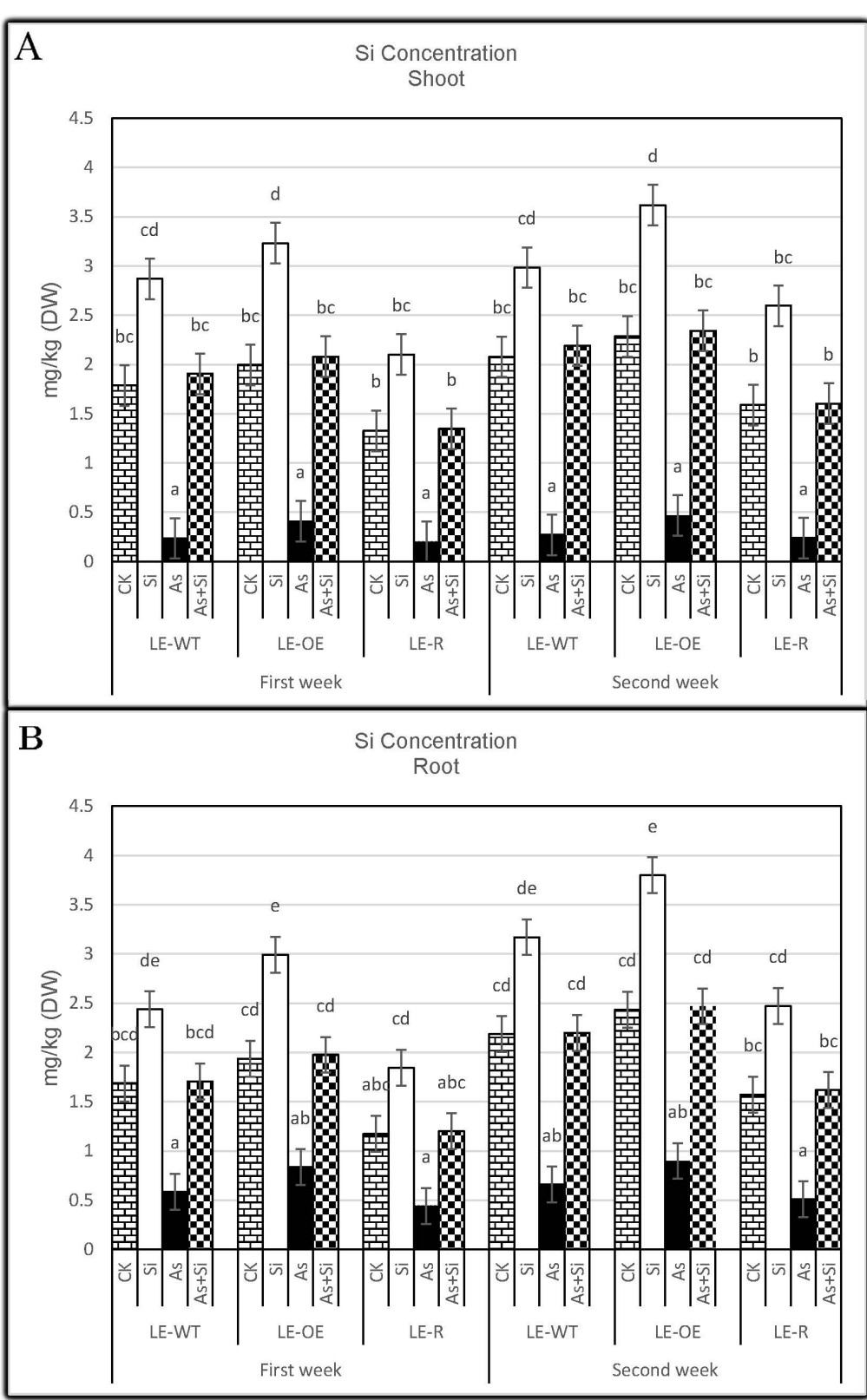

**Figure 4.** Si concentration in shoots (**A**) and roots (**B**) of LE-WT, LE-OE, and LE-R first and second weeks after adding different treatments. The different letters significantly showed differences ($p \leq 0.05$) between treatments, respectively (DW: dry weight).

Adding Si only to the culture medium increased Si concentration in the first and second weeks after treatment, and the highest Si concentration was seen in LE-OE line shoots. Si concentration in LE-OE shoots increased by 61.98% in the first week after treatment but was 58.43% higher than CK in the second week ($p \leq 0.05$).

One and two weeks after As treatments, the concentration of Si in rice seedlings was significantly different compared with CK ($p \leq 0.05$). The lowest Si concentration was seen in LE-R in the first and second weeks, which showed a decrease of 87.93% and 85.08%, respectively, compared to the control. Si concentration in seedlings treated with both As and Si was higher than seedlings treated with As alone. The highest Si concentration was observed in shoots of LE-OE in the first and second weeks, showing a 4.07% and 2.45% increase compared to the control ($p \leq 0.05$), respectively (Figure 4A).

Si concentration decreased compared to control after adding As to the culture medium, and the lowest concentration of Si was observed in the first and second weeks in LE-R roots (Figure 4B). This decrease in Si concentration was 59.45% and 67.60% compared to CK ($p \leq 0.05$). Rice seedlings treated with Si showed increased Si concentration, and the greatest Si concentration was observed in LE-OE roots every two weeks, with a 54.62% and 56.16% increase compared to CK ($p \leq 0.05$), respectively. Si concentration in seedlings treated with As + Si was significantly different from CK, and the lowest Si concentration, both in the first and second weeks, was observed in LE-R line roots, with 2.40% and 2.9% decreases compared to CK ($p \leq 0.05$), respectively.

### 3.3. The Role of Si and As Treatments on Dry Weight of Rice Shoots and Roots

The present study on different treatments showed that the addition of As treatment can significantly reduce the dry weight of different rice roots and shoots every two weeks, and the LE-WT line showed a more considerable decrease in dry weight than LE-OE and LE-R, but the dry weight in the LE-OE line increased further with the addition of different Si treatments. For example, when we added Si, the dry weight of shoots and roots increased in the first and second weeks, and the dry weight was 1.508, 1.912 mg and 0.538, 0.613 mg, respectively, but in the LE-WT line was 1.139, 1.56 mg, and 0.429, 0.497 mg, respectively. Under 30 μM As the same results have been shown in the shoots and roots of LE-WT, LE-R, and LE-OE (Figure 5A,B; Table 2), when we added As the tolerance of LE-OE and LE-R lines in the presence of Si was higher than LE-WT. Silicon treatments can reduce arsenic uptake, and translocation in roots and shoots indicates the role of the *Lsi1* gene in increasing antagonism between As and Si in rice.

### 3.4. The Role of Si and As Treatments on Soluble Protein Contents of Rice Shoots and Roots

Soluble protein content was significantly increased on the seedlings treated with Si, and in both weeks, the highest amount was observed in the LE-OE shoots with a 10.01% and 11.34% increase, compared with CK ($p \leq 0.05$), respectively. Compared to the control, soluble protein contents decreased after adding As to the culture solution ($p \leq 0.05$), and the lowest amount was observed in shoots of LE-WT in the first and second weeks. These decreases were 22.80% and 12.34% compared to CK, respectively (Figure 6A). The soluble protein contents of As + Si treated rice decreased compared to CK ($p \leq 0.05$), and the lowest soluble protein contents were seen in the LE-WT shoots in the first and second weeks (11.25% and 6.66% reduction, respectively).

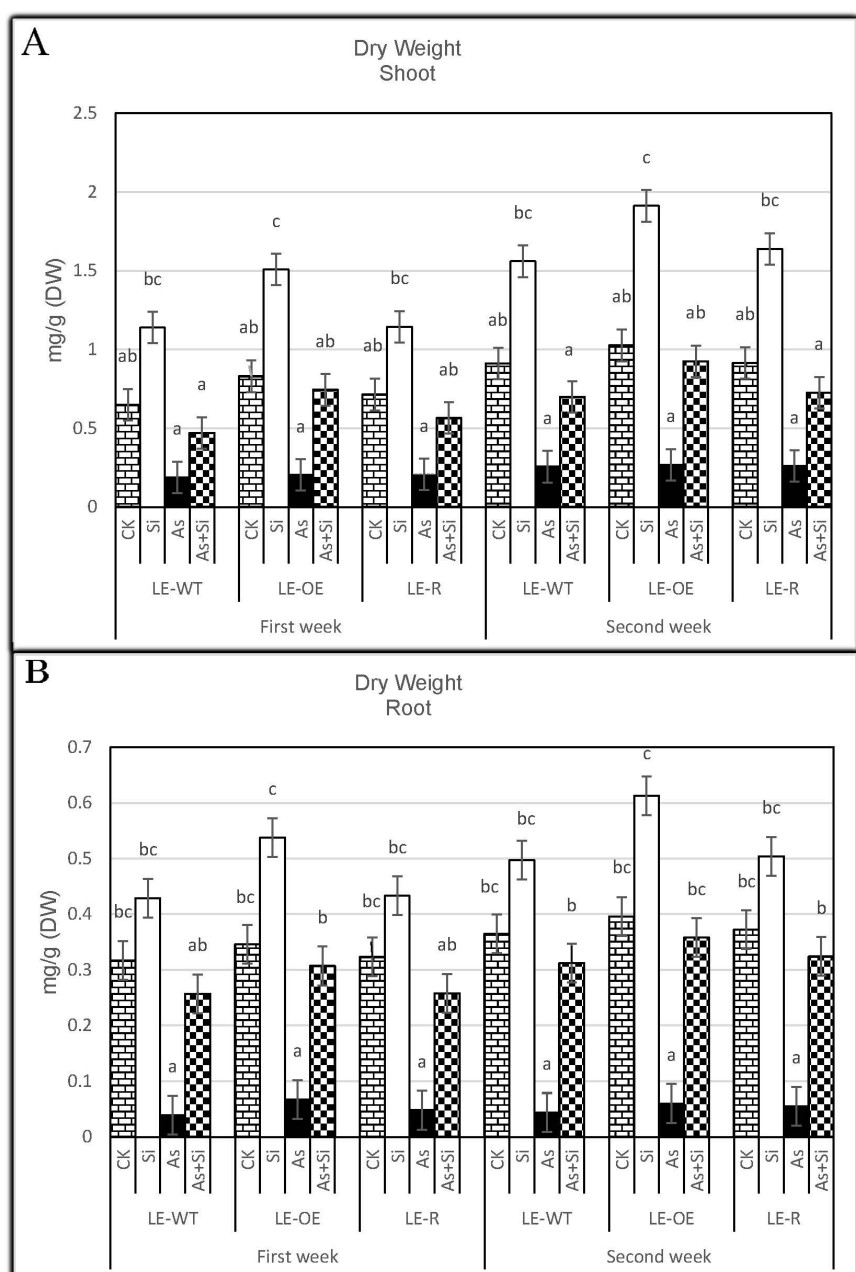

**Figure 5.** Dry weight in shoots (**A**) and roots (**B**) of LE-WT, LE-OE, and LE-R first and second weeks after adding different treatments. The different letters on the bars significantly showed differences ($p \leq 0.05$) between treatments, respectively.

Si treatment showed a significant difference in soluble protein contents in different rice roots compared to CK ($p \leq 0.05$), and the highest increase in the first and second weeks after exposing to Si treatment was observed in LE-R roots, which represented 22.03% and 7.31% increase compared to CK, respectively. Our results also showed that soluble protein contents in different rice lines roots decreased compared to CK at the time of arsenic exposure ($p \leq 0.05$). The most considerable decrease was observed in LE-WT roots, both with 47.42% and 44.15% decrease compared to CK in the first and second weeks, respectively (Figure 6B). Decreased soluble protein content was also observed in the roots of seedlings treated with As + Si in both exposure periods. The most significant decrease was in LE-WT roots in the first and second weeks, representing 37.92% and 29.32% decreased compared to CK.

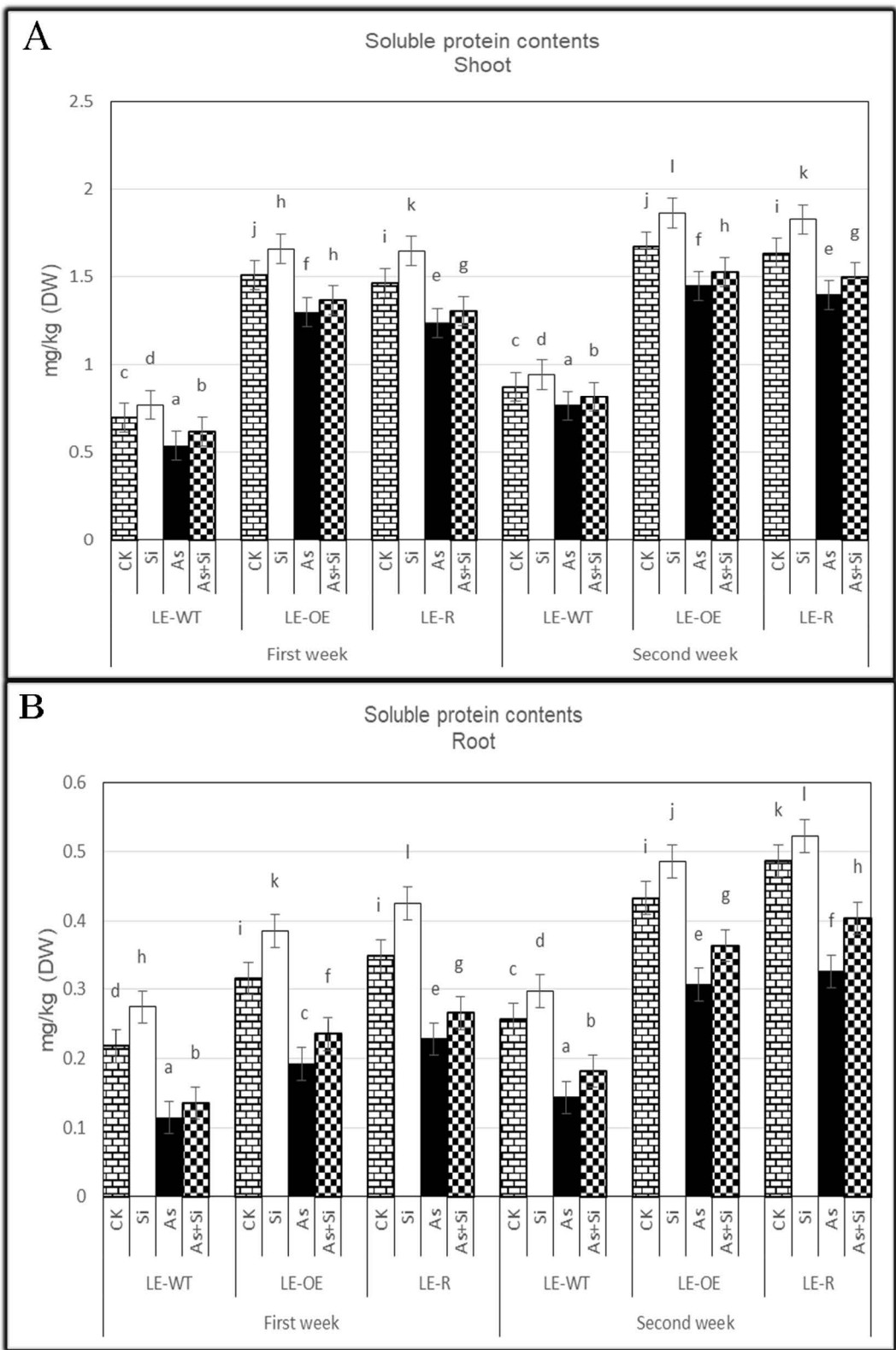

**Figure 6.** Soluble protein contents in shoots (**A**) and roots (**B**) of LE-WT, LE-OE, and LE-R first and second weeks after adding different treatments. The different letters significantly showed differences ($p \leq 0.05$) between treatments, respectively (FW: fresh weight).

### 3.5. Transcriptomic Analyses

To gain global insight, we used Illumina high-throughput sequencing related to the role of As and Si on Oxidative enzymes and non-oxidative enzymes, the seedlings of rice treated with 30 µM As + 0.70 mM Si. Three biological replicates were designed, and Pearson's correlation. The coefficient was calculated to evaluate their reproducibility. By using DESeq, the DEGs were known between the CKs and treatments. Differentially expressed genes (down-regulated and up-regulated genes) are in Figure 7.

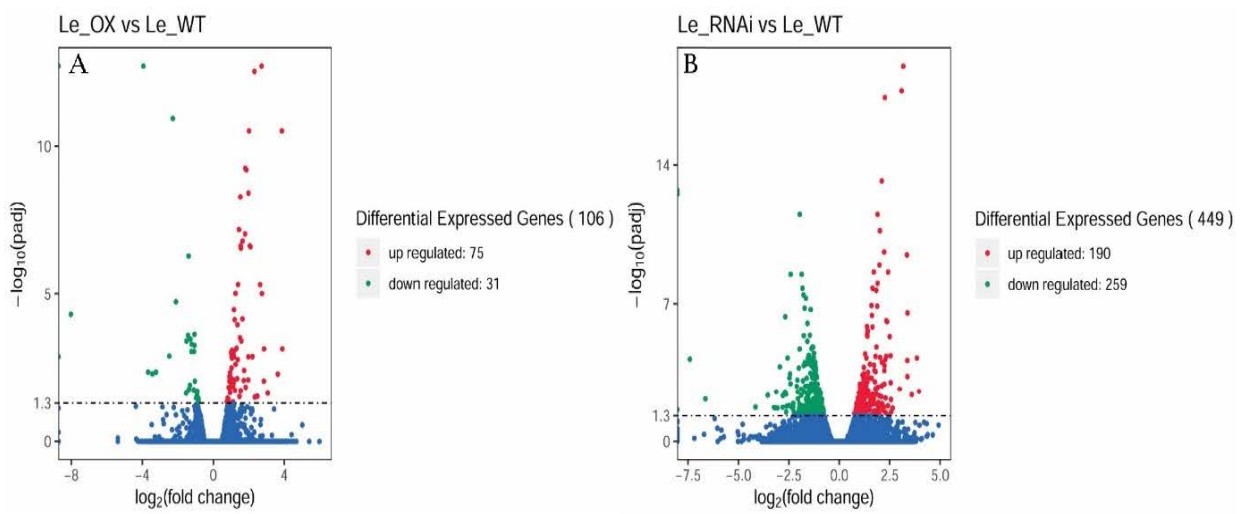

**Figure 7.** Differentially expressed genes LE-OE (**A**) and LE-R (**B**) lines (total, up-regulated, and down-regulated genes).

The list of background queries comprised 106 genes for the LE-OE line and 449 genes for the LE-R line (Figure 7A,B). However, the amount of up-regulated genes was 75 and 190, respectively. We used ViPTree to classify GO on expressed genes to understand the function of As-affected genes and GO enrichment terms of the cellular component, biological process, and molecular function categories. In the biological process classification (GO: 0008150), the enriched DEGs were mainly associated with the metabolic process (GO: 0008152), and in the molecular function classification (GO: 0008150), differentially expressed genes enrichment was mainly related to the metabolic process (GO: 0008152) and catalytic activity (GO: 0003824). In the cellular component classification (GO: 0005575), enriched differentially expressed genes were mainly associated with the cell (GO: 0005623) and cell part (GO: 0044464) (Figure 8).

Among the differentially expressed genes of the LE-OE line and LE-R lines 31 and 259, annotated down-regulated genes were identified. GO enrichment terms of the biological process and molecular function categories. Biological process classification (GO: 0008150), enriched differential expressed genes were mainly associated with the cellular process (GO: 0009987), metabolic process (GO: 0008152), and response to the stimulus (GO: 0050896), but in molecular function (GO: 0003674) enriched differentially expressed genes were mainly associated with the nutrient reservoir activity (GO: 0045735), catalytic activity (GO: 0003824) and binding (GO: 0005488) (Figure 8).

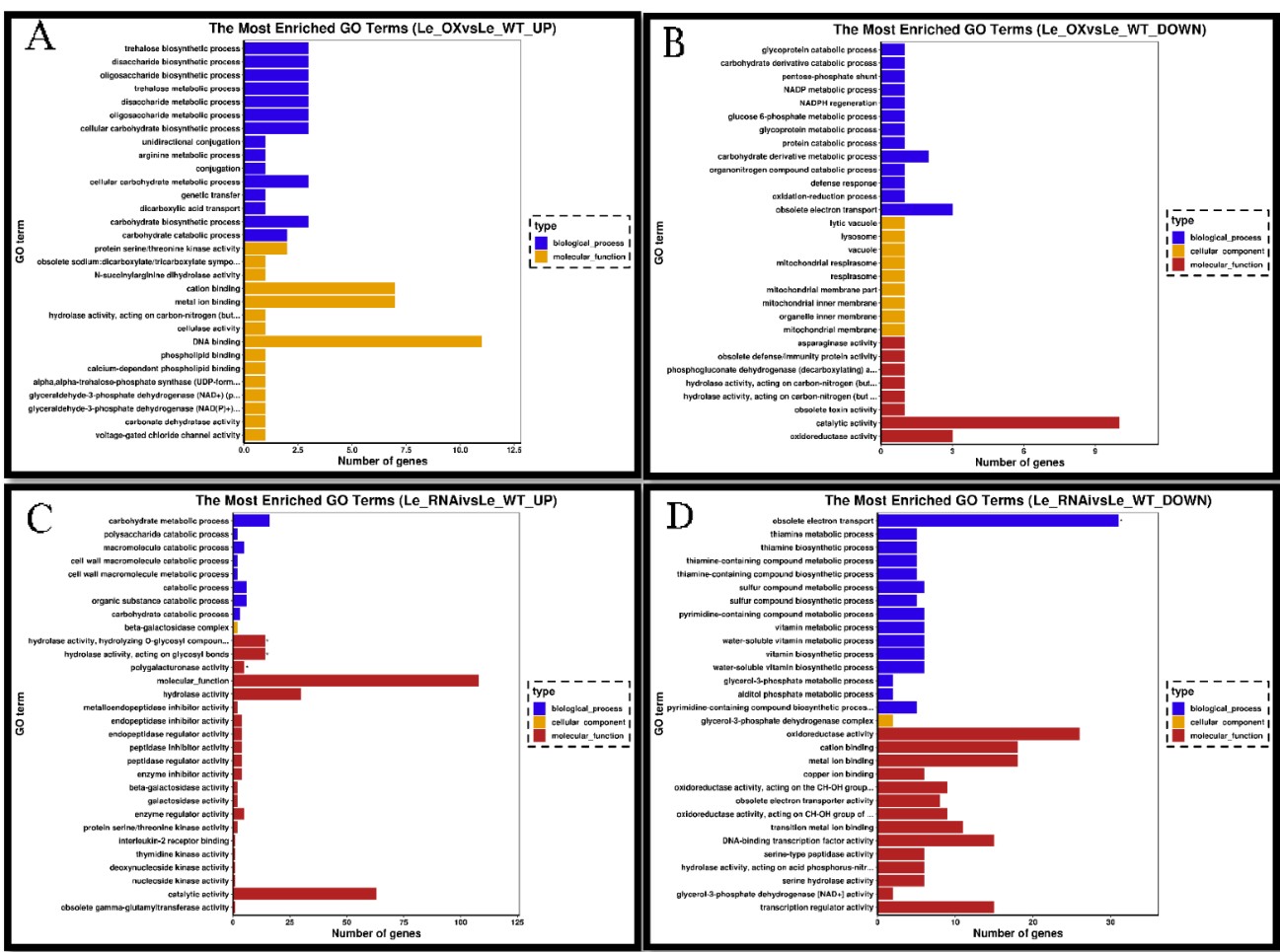

**Figure 8.** GO classification of LE-OE up-regulated (**A**) and down-regulated (**B**) differentially expressed genes, and LE-R up-regulated (**C**) and down-regulated (**D**) differentially expressed genes.

KEGG enrichment analysis showed that in LE-OE line roots, DEGs in plant hormone signal transduction included four genes, three down-regulated and one up-regulated, DEGs in metabolic pathways included six genes, five up-regulated and one down-regulated. Differentially expressed genes in starch and sucrose metabolism included two genes, two up-regulated and zero down-regulated. DEGs in Biosynthesis of secondary metabolites included two genes, two up-regulated and zero down-regulated (Figure 9), and in LE-R roots, DEGs are enriched in plant hormone signal transduction including seven genes, one up-regulated and six down-regulated. DEGs in metabolic pathways contained 35 genes, 21 down-regulated and 14 up-regulated. Differentially expressed genes in Biosynthesis of secondary metabolites included 24 genes, 10 up-regulated and 14 down-regulated. DEGs in glutathione metabolism included six genes, one up-regulated and five down-regulated. It has been observed that the factors that cause rice resistance to As are regulated by the expression patterns of these genes (Figure 9).

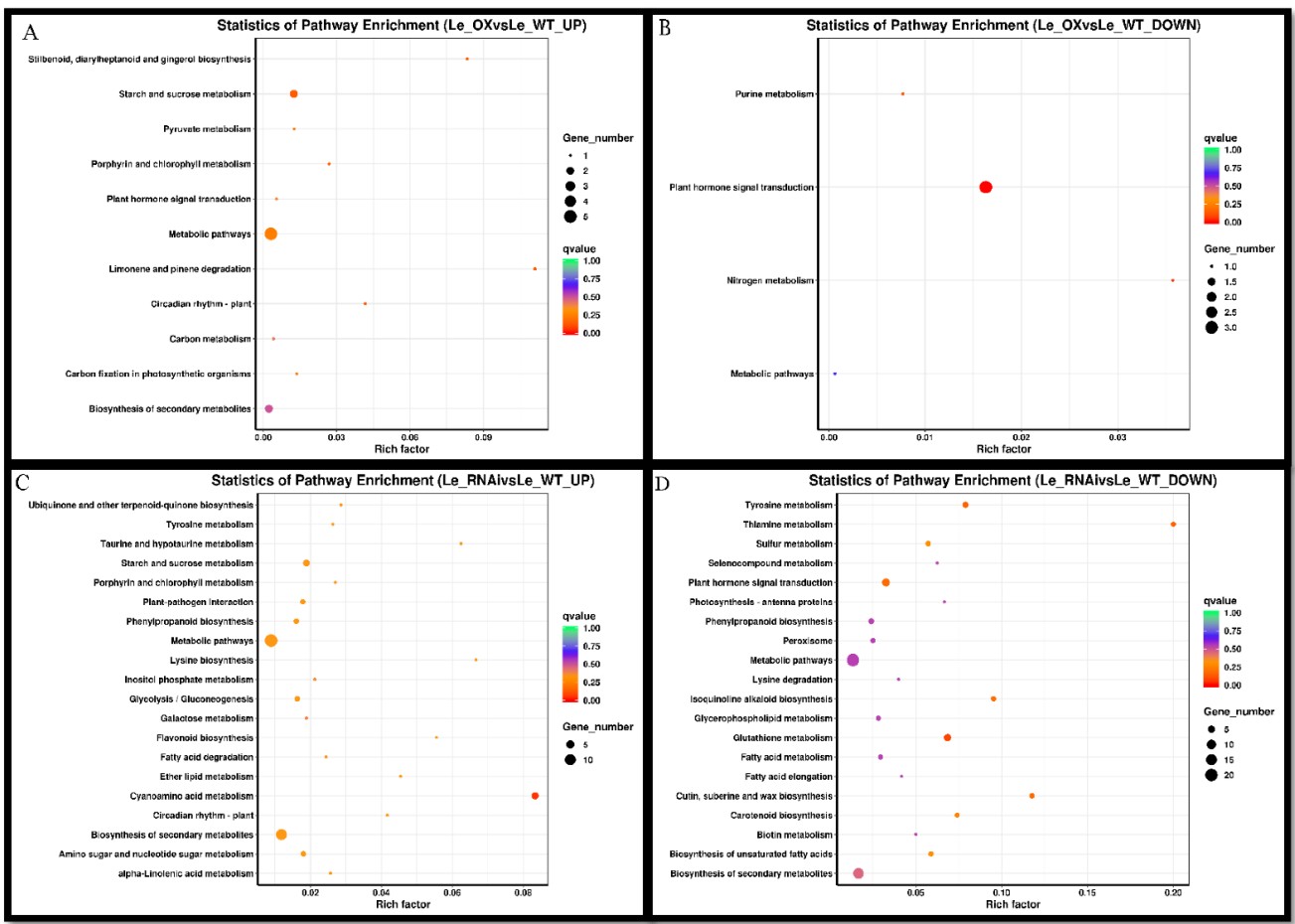

**Figure 9.** The Kyoto Encyclopedia of Genes and Genomes (KEGG) pathway enrichment scatter plot of up-regulated (**A**) and down-regulated (**B**) of LE-OE differentially expressed genes and up-regulated (**C**) and down-regulated (**D**) of LE-R differentially expressed genes.

*3.6. Expression of Some Genes in Different Varieties of Rice Exposed to Si and As Treatments*

Calcium-binding EGF domain-containing protein (LOC_Os10g10130) regulates calcium ion binding and ATP binding in rice. LOC_Os10g10130 expression in LE-OE and LE-R lines was highly up-regulated (Figure 10A) in the presence of 30 µM As + 0.70 mM Si. Si and As increased the relative mRNA of LOC_Os10g10130 in LE-OE and LE-R lines compare to CK (15.40-fold and 29.44-fold, respectively).

The effect of 30 µM As + 0.70 mM Si treatment on ATG8D expression in LE-R line was slightly down-regulated, 0.73-fold, respectively (Figure 10B). However, using qRT-PCR observed that ATG8D expression in the LE-OE line was up-regulated compared to CK (1.13-fold). ATG8D gene regulates the cellular response to nitrogen starvation, Protein transport, transport, and Ubl conjugation pathway in rice.

By using Quantitative RT-PCR, the mRNA expressions of Os10g0530500 that regulates glutathione transferase activity in rice were investigated under 30 µM As + 0.70 mM Si treatments. The results showed that the expression of Os10g0530500 in LE-OE and LE-R lines was up-regulated compared to CK, which was 1.63-fold and 4.33-fold, respectively (Figure 10C).

The relative mRNAs of Os05g0240200 (regulates Plant defense, ADP binding, glutathione transferase activity) in LE-OE and LE-R lines were highly up-regulated compared to CK (1.90-fold and 1.79-fold, respectively) in 30 µM As + 0.70 mM Si treatments (Figure 10D).

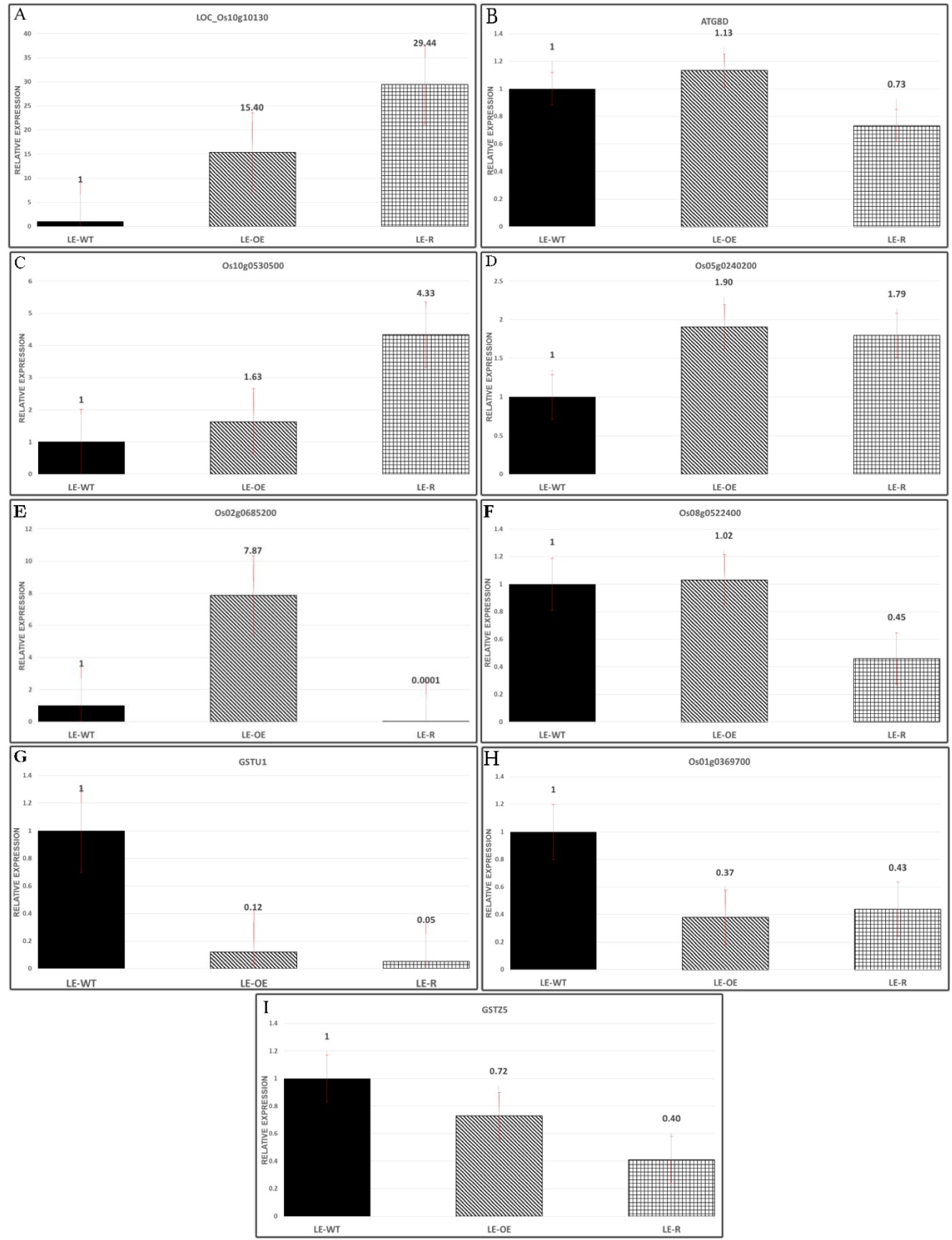

**Figure 10.** Relative expression levels of LOC_Os10g10130 (**A**), ATG8D (**B**), Os10g0530500 (**C**), Os05g0240200 (**D**), Os02g0685200 (**E**) and Os08g0522400 (**F**), GSTU1 (**G**), Os01g0369700 (**H**), GSTZ5 (**I**) RNA transcript in LE-OE, LE-R roots.

Os02g0685200 expression in LE-OE line was highly up-regulated (Figure 10E), Si, and As increased the relative mRNA of Os02g0685200 compared to CK (15.40 fold), but it was slightly down-regulated in LE-R (0.0001 fold), in the attendance of 30 μM As + 0.70 mM Si.

Os08g0522400 regulates peroxidase activity and oxidative stress in rice. Os08g0522400 expression in the presence of 30 μM As + 0.70 mM Si in LE-OE line was increased (Figure 10F), but Os08g0522400 expression in LE-R line was slightly down-regulated (0.45-fold).

The effect of 30 μM As + 0.70 mM Si treatment on GSTU1, Os01g0369700, and GSTZ5 expression in LE-OE and LE-R lines were slightly down-regulated (Figure 10G–I).

## 4. Discussion

The present study compared arsenic and silicon antagonism on the physiological and molecular properties of *Lsi1* transgenic and wild-type Lemont rice lines. *Lsi1* is a capable transporter found mainly in cells of mature plant root regions and affects the accumulation of As (III) and Si in plant tissues [20,32]. Research has shown that *Lsi1*-inhibitor or *Lsi1*-overexpressing transgenic rice is more resistant to cadmium toxicity in the presence of silicon. The increase in cadmium stress resistance depends not only on the amount of Si in the culture medium but also on the expression of *Lsi1* in plants. [33].

Si is very valuable for plant growth, especially in stressful conditions [34], and this is an essential element in plants to reduce the toxicity of various metals such as Zn, As, and Cd. It can also reduce the absorption and transport of metals in plants, especially rice. Many studies have reported that Si application improves the tolerance of toxic metals in various plants, which reduces the absorption of heavy metals in the roots and their transfer to the shoots, and ultimately their toxicity in plant tissues [14,35]. It should be noted that research on Indica rice has also shown that Lsi1 plays a key role in Si accumulation in transgenic rice lines. [36]. According to research, Si and As have the same transport routes in rice [20], and Si application reduced its adsorption due to direct competition between Si and As during adsorption [20,37].

Similar research has shown that the use of Si reduces the concentration of As in various organs of rice plants such as straw, stamen, and leaves by half [13,14]. Additionally, the phenotypic difference of rice causes Si accumulation in different organs to be different [38], and these results are similar to our experiment.

Some studies have shown that various rice genotypes have different potential for As uptake and accumulation, and adding the same level of As to different rice genotypes causes the different As accumulation [3,16]. Similar greenhouse experiments also showed changes in As content in pots and made a significant difference in As accumulation of different types of rice and different parts of plants [2]. Studies have also shown that suppression of the *Lsi1* gene reduces arsenic accumulation in rice roots and shoots [39], and similar results were found in our study.

The role of different stresses on the amount of soluble protein has been investigated less than antioxidant enzymes. In this study, the addition of Si solution to As significantly increased the soluble protein content of different rice strains compared to arsenic treatment alone. It suggests that Si plays a vital role in neutralizing ROS toxicity, reducing excessive stress, and ultimately reducing oxidative damage, reducing protein degradation, improving protein synthesis, and increasing protein metabolism under stress [40,41]. These results are consistent with other research observations that show the effect of Si on increasing soluble protein in plants under stress, which is also dependent on Si concentration [42,43]. This study found that the soluble protein of rice seedlings treated with As was reduced, indicating that damaged seedlings and protein metabolism were disrupted by arsenic stress. Decreases in the soluble protein levels may be due to the following reasons: As reacts with sulfhydryl groups of proteins that cause the damage in rice tissues, potent inhibition of rice growth and photosynthetic activity, more oxidative damage, which inhibits protein synthesis, increases the rate of destruction by disrupting the membrane system, reduces the level of As (V) to As (III) in cells responsible for protein damage via oxidation of thiol

groups [41,42]. Decreased protein content in rice and other plants under various stresses has been observed [40,43,44].

The effects of the *Lsi1* gene are not completely limited to increasing Si in roots and shoots of *Lsi1* transgenic rice lines, but different studies in hydroponic and in vitro conditions have shown that adding Si to the culture medium significantly increased the root and shoot dry weight and fresh weight, root length, chlorophyll content of *Lsi1* transgenic rice lines more than their wild type [32,36].

Heavy metal pollution is an increasing environmental problem, which is very toxic and dangerous to plants. Much research has been done to establish the molecular, physiological, and genetic basis of As tolerance in plants [22]. However, uptake, transport, and accumulation mechanism of As in plants and the transcriptional regulation of this process is unknown. Therefore, there is an urgent need to address these questions significantly when As is growing in the rice mediums and abundant in different parts of rice [37]. Prior studies used microarray techniques to check the transcriptional response regulation of As stress in the plant. The Illumina sequencing approach has also been used to determine many differentially expressed genes [22]. We tried to investigate genes and mechanisms related to arsenic stress through GO (gene ontology) enrichment analysis in this experiment. The present study tried to examine arsenic stress on DEGs in different pathways in rice roots. These pathways included glutathione metabolism, biosynthesis of amino acids, RNA degradation, biosynthesis of secondary metabolites, plant hormone signal transduction, metabolic pathways, etc. Different studies have also emphasized the effect of these mechanisms in reducing arsenic stress [22,37].

Our study investigated the differences in susceptibility among three different rice cultivars to As. This study also tried to determine GST genes and other antioxidative enzyme genes' effect on reducing arsenic contamination. The number of genes related to glutathione S-transferase (GST) was significantly increased. These intracellular proteins (GST) are found in prokaryotes, eukaryotes, and aerobics, supporting cells against other chemical toxicity [45].

GSTs are the essential enzymatic systems for phase II metabolism, which enter the plant system as the primary system of exogenous chemical metabolizing enzymes. GSTs detoxify heavy metals and herbicides by catalyzing their binding to glutathione [46]. GST genes had different expressions in different varieties, which indicates that different GST genes have different expressions in several varieties under stress conditions. These results align with studies on other plants and under different stresses [47,48].

As ROS due to abiotic stresses was increased, the related genes' expression also grew to improve the plant defence situation [49]. The decrease and increase in antioxidative enzymes due to different stress cause the decrease and increase in $H_2O_2$ in the plants; thus, it induces antioxidant-related gene expression. These genes' expression is also different in diverse lines of a plant [50]. In our study, the genes related to antioxidative enzymes also showed similar results.

Rice is known as an efficient Si collector among all crops. It has been reported that exposure to Si contributes to the plant's growth under metal stress, such as As. Different studies have been conducted to clarify the role of Si in reducing heavy metal toxicity at the gene level. The Si-mediated mechanisms in reducing heavy metal toxicity are not understood at the molecular and genetic levels. Still, they have shown that Si application in rice treated with heavy metals significantly activated some genes [51,52]. The plant cells have different transporters, facilitating arsenic uptake and accumulation and other elements present in the plants. These include ATP binding cassette transporter, aquaglyceroporins, phosphate transporters, and nitrogen transporters [53]. It is essential to decrease unwanted elements such as heavy metals by regulating these necessary transporters, which reduces the toxic elements in the plant [54]. The present study showed that gene expression involved in elemental transport was up-regulated when some lines were exposed to Si and As.

## 5. Conclusions

The experiment compared wild-type rice with its transgenic lines, and it has shown a direct relationship between the concentration of As and Si in the culture medium and their accumulation in rice. Moreover, increasing Si had an active role in reducing arsenic stress, and this effect was more significant in the LE-OE line than in other varieties. The study also showed that adding As to the culture medium in all lines reduced the soluble protein in both root and shoot, which the highest decrease in soluble protein was observed in LE-WT. However, adding Si has gained the soluble protein in all rice and reduced the effect of arsenic on soluble protein in the shoots and roots. Gene ontology (GO) enrichment analysis of roots treated by As and Si showed 106 DEGs in the LE-OE line and 449 DEGs in the LE-R line, and the different genes expressed were belonged to different gene families indicating the complex mechanism of rice to respond to arsenic stress conditions.

The study results showed that the genes that had a role in antioxidant enzymes and element transport in rice were up-regulated, and this highlights the significant role of these genes in reducing arsenic contamination in transgenic Lemont rice lines. Therefore, we suggest that future experiments focus on the effect of different levels of Si and As on gene expression. Since transgenic rice is more resistant to arsenic toxicity in the presence of silicon, planting transgenic rice on farms can improve yield and food safety. In addition, if Si fertilizers are added to the culture medium, the best option for rice cultivation is the LE-OE line because it has low As accumulation and high soluble protein.

**Author Contributions:** M.R.B., W.L., C.F. designed the experiment. M.R.B. and Y.J. performed most of the experiments. M.R.B. analyzed data and wrote the manuscript. W.L. and C.F. edited the manuscript. All authors read and agreed to the published version of the manuscript.

**Funding:** This work was supported by the Outstanding Youth Scientific Fund of Fujian Agriculture and Forestry University (Grant No. xjq201805). The funders had no role in study design, data collection, analysis, data interpretation, or manuscript writing.

**Institutional Review Board Statement:** Not applicable.

**Informed Consent Statement:** Not applicable.

**Data Availability Statement:** All data generated during this study are included in this published article and its supplementary information files, and the raw data used or analyzed during the current study are available from the corresponding author on reasonable request.

**Acknowledgments:** We thank Gabor Pozsgai and Jiang Yuhang for their constructive comments and suggestions to complete this paper.

**Conflicts of Interest:** The authors declare that they have no competing interest.

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
