# Peer review of "Silicon Modulates Molecular and Physiological Activities in Lsi1 Transgenic and Wild Lemont Rice Seedlings under Arsenic Stress"

_agronomy, doi:10.3390/agronomy11081532_

Round 1

Reviewer 1 Report

In the manuscript entitled "Silicon modulates molecular and physiological activities in Lsi1 transgenic and wild rice seedlings under arsenic stress" the authors investigated the As and Si contents, dry weight, soluble protein content, and performed transcriptome analysis on treated rice lines. The research work is well-planed and performed, the methods were described in detail. In general, the manuscript is written in good manner and order.

Further questions and comments:

  • Line #36: The citation “Bhattacharya et al., 2013” remained in the text, and it is Nr. 16 in the reference list. Therefore, the reference list needs to be corrected.
  • Line #162-178: In Paragraph 2.6. very difficult to see through the selected genes and the designed primers. I suggest preparing a Table with gene codes, protein names and primer sequences.
  • Line #217 and 219: “≤” sign is missing between p and 0.05.
  • Table 1: There are two typos in the column of “Parameter” (As concentration)
  • Figure 2/B: The diagram is slipped, therefore the y axis title is not visible.
  • Figure 3: Please, double-check the y axis titles. I think so the unit is not correct, should be in gram.
  • Figure 4. Please, double-check the y axis titles. I think so the unit is not correct.
  • Figure 8. The Figure is designed very badly. The texts are not readable, while the graph columns are unnecessarily too wide. I suggest, to prepare the Figure again. For example, in 3*3 arrangement.
  • What is the basis of the selection of the genes? For example, from many GST genes why the authors chose GSTU1?
  • What can be the reason that the soluble protein contents and the expression level of some genes (GSTU1, Os01g0,69700, LOC_Os10g10130) in Lsi-RNAi and LSi-OE lines changing similarly?

Author Response

Dear Reviewer,

Thanks for your helpful comments.

You can find the explanation of the modification below:

  • Reread all the manuscript and modified grammar mistakes.
  • Line 36, the citation has been modified, and also all references have been reviewed and modified.
  • In Materials and Methods, all descriptions for Transcriptomic analysis and quantitative RT-PCR have been completely changed, and all primers are included in a table, and two more Figures are added to make the materials and methods clearer.
  • The mistakes of Table 1 ( 2 . new) have been corrected.
  • Figure 2/B (4/B. New) has been corrected.
  • Figure 3 (5/ new) has been modified and also inside the text.
  • We made Figure 8 (10. New) bigger as much as possible.
  • We selected the genes according to the RNA seq result, selected the more significant genes, and QPCR confirmed the RNAseq result.
  • The soluble protein content is not an independent result to evaluate plant resistance; it calculates the antioxidant activity and combined it to explain the plant resistance to stress. The arsenic stress may suppress this gene expression because the OE and RNAi lines rice root have higher arsenic contents; the influence of arsenic on this gene expression may be greater than silicon influence, but the underlying mechanism needs additional study.

Best Regards

Reviewer 2 Report

The present manuscript entitled “Silicon modulates molecular and physiological activities in 2 Lsi1 transgenic and wild rice seedlings under arsenic stress” demonstrated that transgenic cultivars were more resistant to arsenic than wild-type, especially when silicon was added to the culture medium. The manuscript contains significant data and presents the results in detail; moreover, there are some suggestions for further improvement, please find below.

1-      There are several grammar mistakes in language, for example, the first line of the abstract “Arsenic is one of the most dangerous metalloids” may be written as “Arsenic is one of the most dangerous metalloid”.

2-      Line 50-52 “Rice is an essential crop in the world that provides 40% of people's food. It provides two-thirds of the calories needed by two billion people in Asia and is also the main source of protein for this population”, please provide reference proving the main source of protein.

3-      Line 109-110 “Transcriptomic analyses and quantitative RT-PCR” methodology should be explained in detail.

4-      Line 85 “Lsi1 Lemont transgenic lines” were tested for molecular response and published earlier (reference 23), please explain how different the present study and data are? or is it just a reconfirmation?

5-      2.4. Soluble protein contents measurement- method should be in detail.

6-      Line 145 “We used one µg for extracting RNA for the first-strand” please correct this sentence and mention which RNA was used for cDNA synthesis?

7-      Line 158 “The synthesis of cDNA was done according to the instructions of the manufacturer” which manufacturer?

8-      Line 161 “GST-related genes' relative transcript level enhanced rice plant resistance to As stress determined via Quantitative RT-PCR in three replications.” Please rewrite this sentence.

9-      2.6. Quantitative RT-PCR- Primer sequences may be presented in a table form with details of Tm and may be replaced in the supplementary material. Additionally, please mention which were the reference genes for Quantitative RT-PCR and how many reference genes have been used in the present study? Please provide the melt curve data if only one reference gene has been used.

10-  Line 632-634 “The present study is the first study to compare arsenic and silicon antagonism on the physiological and molecular properties of Lsi1 transgenic and wild-type Lemont rice lines. Lsi1 is a capable transporter, found mainly in cells of mature plant root regions, that affects the accumulation of As (III) and Si in plant tissues [19, 30]” Please avoid writing that it is the first study.

11-  Conclusion should be shorter and clearer to convey the key findings and their application for a better understanding of the readers from diverse areas.

Author Response

Dear Reviewer,

Thanks for your helpful comments.

You can find the explanation of the modification below:

1- Reread all the manuscript and modified grammar mistakes.

2- References added to the text (Reference 10).

3- The details were added to the manuscript.

Total RNA from rice root was extracted by using TRIzol (Invitrogen), according to the manufacturer’s instructions. RNA quality was measured on an Agilent 2100 Bioanalyzer system (Agilent Technologies, USA). Total RNA (0.75 μg) from each test sample was treated with oligo (dT) beads to enrich mRNA and fragmented using fragmentation buffer (New England Biolabs Ltd., UK). mRNA was reverse-transcribed into cDNA using random hexamers, and double-stranded cDNA (ds cDNA) was synthesized and purified. After filling in the cohesive ends to yield blunt ends, ds cDNA was combined with poly (A) and ligated to Illumina sequencing adapters (Illumina Inc., USA). The ligation products were size-selected by AMpure XP beads (Beckman Coulter Life Science)and amplified by PCR, and cDNA libraries were sequenced on an Illumina Hiseq PE150 Allwegene Tech. (China).

4- The present study, “Lsi1 Lemont transgenic lines,” was tested for molecular response to Arsenic stress, but previous studies tested for molecular response to ultraviolet radiation.

5- The details added to the Soluble protein contents measurement- method.

6- The correction was done, and details were added to the manuscript same as below:

Total RNA (0.75 μg) from each test sample was treated with oligo (dT) beads to enrich mRNA and fragmented using fragmentation buffer (New England Biolabs Ltd., UK). mRNA was reverse-transcribed into cDNA using random hexamers, and double-stranded cDNA (ds cDNA) was synthesized and purified using a QIAquick PCR extraction kit (QIAGEN Inc., USA).

7- The details added to the manuscript same as below:

according to the instructions of the TransScript One-Step gDNA Removal and cDNA Synthesis SuperMix manufacturer (TransGen Biotech Co., Ltd. Beijing)

8- The sentence was rewritten.

‘Relative levels of GST-related genes were increased rice plant resistance to arsenic stress, and they were determined through quantitative RT-PCR in three replications.’

9- Two Figures and one table (Fig 1, 2 and Table 1) was added and added description same as below:

The quantitative RT-PCR reaction system was prepared using TransStart Tip Green qPCR SuperMix and an Eppendorf realplex4 instrument. The reaction process was as follows: predenaturation at 94 °C for 30 s, denaturation at 94 °C for 5 s, annealing at 53 °C for 15 s, extension at 72 °C for 10s; 42 cycles. When the amplification was finished, the melting curve analysis was conducted, and the specificity of the product was determined based on the melting curve. Each candidate mRNA was set with four independent replicates. The relative expression of the gene was calculated by the 2-△△Ct method with the threshold cycle values (Ct) of each candidate mRNA in both the control and test samples

10- The sentence was modified.

11- The Conclusion was modified and made shorter.

Round 2

Reviewer 1 Report

The authors tried to correct the manuscript according to the suggestions. But it was not completely successful.

  • New Figure 5: Why the dry weight unit in y axis title is "mg/g (DW)" ? What does it mean? How did they measure the dry weight to get this unit?
  • New Figure 6: In the legend of the figure the authors wrote "FW: fresh weight", however on the graphs y axis title there is still "DW" so dry weight. Need to correct the graphs.
  • New Figure 10: there was a misunderstanding. I suggest to make ONLY the text and titles bigger on the graph, not the whole graphs. The colums are unnecessarily wide, taking the space from the text, what is still unreadable.
  • The answer for the question is fair enough, but it is still strange, that in an overexpressing and RNA interference line (because of Arsenic level) similar effects can be detected in gene expression level.

Reviewer 2 Report

The authors addressed all my remarks. The current version of MS can be accepted.